# A generic blood banking and transfusion process-oriented architecture for virtual organizations

Anwar Rjoop[1]*, Shaima' Elhaj[2], Dina Tbaishat[2,3,4], Yousra Odeh[2,5], Asem Mansour[2], Mohammed Odeh[2,6,7]*

1 Department of Pathology and Microbiology, Faculty of Medicine, Jordan University of Science and Technology, Irbid, Jordan, 2 Center for Cancer Care Informatics Research, King Hussain Cancer Center (KHCC), Amman, Jordan, 3 College of Technological Innovation, Zayed University, Dubai, UAE, 4 Information Science Department, University of Jordan, Amman, Jordan, 5 Software Engineering Department, Faculty of Science and Information Technology, Al Zaytoonah University of Jordan, Amman, Jordan, 6 College of Arts, Technology and Environment, University of the West of England, Bristol, United Kingdom, 7 Global Academy for Digital Health, United Kingdom

☯ These authors contributed equally to this work.
* aarjoop@just.edu.jo (AR); mohammed.odeh@sky.com (MO)

**Data Availability Statement:** All relevant data are within the manuscript and its Supporting information files.

## Abstract

### Background

Blood banks are an important part of healthcare systems. They embrace critical processes that start with donor recruitment and blood collection, followed by blood processing to produce different types of blood components used in transfusions, blood storage, blood distribution, and transfusion. Blood components must be generated at high quality, preserved safely, and transfused in a timely manner. This can be achieved by operating interrelated processes within a complex network. There is no comprehensive blueprint of Blood Banking and Transfusion (BB&T) processes and their relationships; therefore, this study aims to develop and evaluate a BB&T process architecture using the Riva method.

### Research design

This research adopts a design science research methodology process (DSRM) that aims to create artifacts for the purpose of serving humanity through six phases: identifying problems, identifying solutions and objectives, designing and developing artifacts, demonstrating and evaluating the artifacts, and communicating the work. The adapted DSRM process is used to build a process architecture in the BB&T unit to improve the quality and strategic planning of BB&T processes. Applying the adapted DSRM process generated four increments before the outcomes were communicated as a highly comprehensive BB&T process architecture (BB&TPA) blueprint for virtual organizations. Finally, the generated BB&TPA is tested and validated at a reference hospital.

**Funding:** The author(s) received no specific funding for this work.

**Competing interests:** The authors have declared that no competing interests exist.

## Results

A Riva-based process architecture diagram was successfully developed, acting as a reference model for virtual BB&T organizations. It is a novel output in the domain of BB&T and can also be considered as a reference model to evaluate the existing processes in BB&T real-world units. This assists domain experts in performing gap analysis in their BB&T units and paths for developing BB&T management information systems and can be incorporated in the inspection workflow of accreditation organizations.

## Introduction

Blood banks are mission-critical systems and an important part of healthcare systems. They embrace critical processes that start with donor recruitment and blood collection, followed by blood processing, storage, and distribution to produce different types of blood components used in transfusions. These blood components must be generated at high quality and preserved safely. Technicians, nurses, and physicians face various challenges in managing transfusion complications. Blood shortage is a major concern for stakeholders and has been exacerbated more in middle- and low-income countries [1]. This limits the timely administration of blood units when needed. According to Abderrahman and Saleh, this is a major concern, especially in developing countries such as Jordan and Saudi Arabia [2]. Identifying all processes in the blood bank and linking these processes together is essential for early monitoring of workflow and troubleshooting errors.

Business Process Architecture (BPA) is one of the most important methods utilized in business domains to blueprint a common understanding of running processes and their dynamic interactions from a high-level perspective. BPA can assist stakeholders in identifying process gaps and pitfalls [3]. To minimize risks, attain good blood banking and transfusion clinical and planning practices, and improve the quality of blood banking and transfusion (BB&T) products and services, it is worth looking at the existing processes more closely. Not only this, but we are also looking forward to digitizing this organization and making it virtually available to all stakeholders, namely the BB&T virtual organization.

In the literature, the Riva method has been employed to develop a BPA for various organizations in the healthcare sector, such as cancer care centers and cancer registries [4–9]. "Riva is a method for eliciting, analyzing, and designing the organizational architecture of business processes." [3]. The resulting BPA of the Riva method is based on deriving essential business entities that have an important lifetime for the organization [3, 5, 10, 11].

The remainder of this paper is structured to describe the following: background, research design used to develop Process Architecture PA, evaluation and reflections on BB&T PA, conclusion, and future work.

## Background

The BB&T unit at King Abdullah University Hospital (KAUH) is a division in the pathology and laboratory department that offers transfusion services by handling blood donations and producing various types of blood components such as packed red blood cells, fresh frozen plasma, platelets, and cryoprecipitate. In addition, it provides routine blood tests, such as blood genotyping, blood phenotyping, cross-matching, and direct and indirect Coombs testing [12, 13]. It also provides infectious disease tests and blood component modifications (e.g.,

irradiation and leukoreduction) [13]. These services require the operation and coordination of many activities, starting from the blood bank reception, where a blood donor questionnaire is completed, followed by drawing blood in a donation room. Subsequently, blood storage, processing, and preparation take place in the designated rooms. Finally, a blood transfusion is performed in the inpatient department. The blood bank of KAUH receives approximately 10,000 regular donors annually and performs 193 single-donor procedures.

To understand business processes and their interactions in the big picture, a process architecture diagram is necessary for development and maintenance. In the literature, BPA are referred to as clusters [14], process maps [15, 16], or process landscapes [17–19]. Despite the crucial role of blood banks and transfusion in health systems and services, the literature lacks a blueprint that presents the entire blood banking and transfusion processes and their relationships. Consequently, the development and application of BPA in this field have not yet been explored. The aim of this study is to develop and evaluate a generic Blood Banking and Transfusion Process Architecture (BB&TPA) designed using the Riva method.

## Methods and design

### Ethical considerations

This study was conducted according to the principles expressed in the Declaration of Helsinki; no personal patient information was used, and the outcomes did not have a direct impact on patient treatment. This study was approved by the institutional ethics review board (IRB number 39/139/2021).

**BB&T process architecture development (Research framework design).** The design of this research adopts the Design Science Research Methodology process (DSRM) that aims to create artifacts to serve humanity through six phases: identifying problems, identifying solutions and objectives, designing and developing the artifact, demonstration, evaluating the artifact, and communicating the work [20]. During the entire life cycle of the DSRM process, the phases iterate, overlap, and incrementally deliver the output to adjust the development of an artifact for the best benefit to address a solution to an identified problem [20]. The DSRM has been widely used in many health informatics studies in the literature [4–6]. It has also been used to design a Riva-object-based process architecture and evaluate the effectiveness of its heuristics [4].

The Riva method for identifying an organization's business process architecture comprises seven steps:

1. Identify the organization and its boundaries. This means identifying what we want to look at.

2. Characterize an organization that can be specified abstractly through objectives. This is elaborated by eliciting a set of candidate essential business entities by answering a set of questions, such as "What do we make?", "What services do we offer," and so on [3]. The questions were customized to fit the domains.

3. Find the essential business entities.

4. Identify the units of work.

5. Identify dynamic relationships between units of work.

6. Transform the unit of work diagram into a first-cut process architecture.

7. Transform the first-cut process architecture into a second-cut process architecture.

Step 1 and Step 2: The Riva BPA method has seven steps, as illustrated above: The first and second steps align well with the first and second phases of the adapted DSRM process for this research, related to problem identification and defining objectives; in this case, the problem is specified in the absence of a process architecture in the BB&T unit at KAHU to improve the quality and strategic planning of BB&T processes. This research gap was identified based on the literature. The rest of the Riva method steps were applied in the third, fourth, and fifth phases. As for steps 3–7 of the Riva method, they are considered a form of four iterations over phases 3 to 5 of the adapted DSRM process, generating four increments before the outcomes are communicated and publicized through the concerned stakeholders and publications in phase 6.

The BB&T Process Architecture (PA) is a virtual representation of this unit at KAUH, where its processes evolve with emerging changes in the domain and organization. Therefore, this study aims to develop a highly representative BB&T PA to enact as a lower boundary unit where all processes and their associated resources coordinate to achieve a lower boundary goal. BB&T PA allows similar BB&T units to reuse the resulting BB&T PA to specify whether it is fully or semi-virtual or not. Two research questions (RQ) were formulated to achieve the aims of this study.

RQ1: How to design the BB&T unit at KAUH using the Riva method?

RQ2: How to evaluate the representativeness of BB&T PA that involves processes and their relationships?

Phase 1: Define the Research Problem:

In this phase, the research problem was identified. Based on the literature, a notable absence of process architecture in the BB&T unit at KAUH for the purpose of improving the quality and strategic planning of BB&T processes is recognized.

Phase 2: Define the Objectives of the Solution:

This phase involves identifying the objective of this research: to develop and evaluate a comprehensive BB&T PA using the Riva method to act as a representative reference model for a process-oriented BB&T virtual organization. To enrich the specification of this objective, this phase involves applying the first and second steps of the Riva method [3]. The first step entails agreeing with an organization's business boundary and the scope of interest that the organization is interested in [3]. In this regard, we are interested in examining the blood banking and transfusion domains at KAUH [12, 13]. This indicates the boundary of the anticipated solution, namely, the BB&T PA at KAUH.

According to the Riva method, the second step entails characterizing the BB&T organization. This step enriches the description of the anticipated solution. A list of 14 heuristics in the form of questions should be answered to elicit Candidate Essential Business Entities (CEBEs). The questions were distributed to 20 key stakeholders (S1 Appendix) in the form of a questionnaire (S2 Appendix) in the period from 01.04.2021 to 30.04.2021 (the stakeholder is defined as a person with a vested interest, or stake, in the decision-making and activities of the blood bank and transfusion services). The adult participants provided informed consent. The authors contributed to the 14 heuristic questions by permitting stakeholders to score the degree of confidence for each CEBE, indicating the clarity of each heuristic (yes/no question), and suggesting rephrasing of each heuristic if needed. As the participants were extremely busy, getting them together in a brainstorming session was very difficult; therefore, each heuristic was explained in detail to each participant individually in a face-to-face meeting. The responses were submitted within one week, and as per the participants' feedback, they believed that some

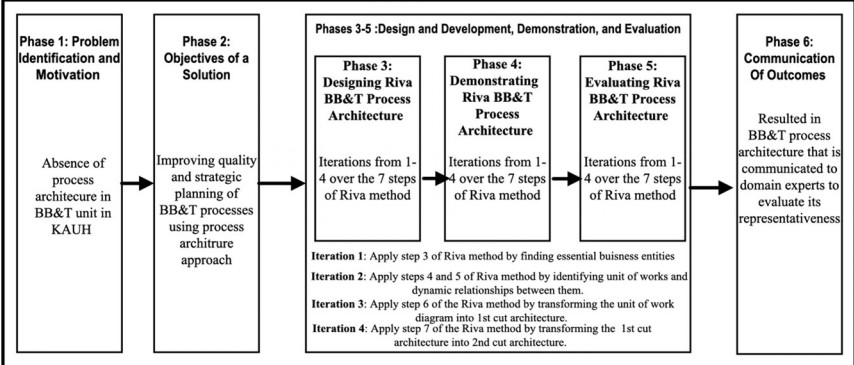

**Fig 1. The BB&T process architecture development research framework.**

open-ended questions required in-depth thinking. The questionnaire resulted in 150 CEBEs that were reduced to 136 after duplicates were removed.

Phases 3 to 5: Design, Development, Demonstration, and Evaluation

The Riva method steps from 3 to 7 are conducted in the form of iterations over phases 3 to 5 of the adapted DSRM process, generating four increments before the outcomes are communicated and publicized through the concerned stakeholders and publications in phase 6. Fig 1 summarizes the phases of the framework.

Step 3: Find the Essential Business Entities

Different filters were used to filter CEBEs into Essential Business Entities (EBEs). For example, one filter requires that "a" or "the" be prefixed to each CEBE in turn, and if the resulting phrase makes sense, then the CEBE is deemed to be an EBE [3]. The final number of EBEs was 80, as listed in Table 1. It is worth mentioning that two meetings were held with domain experts to validate the filtration of the CEBEs into EBEs. In addition, a few EBEs have been renamed for ease of use [3]. For example, an EBE called "manage the transfusion reaction" is renamed to "post-transfusion," which will include all activities happening after the initiation of the transfusion. In addition, the word "manage" will be added in front of the EBEs at some later steps, if needed, according to Riva method instructions [3].

Step 4: Identify the Units of Work (UOW):

In this step, EBEs were filtered into Units of Works (UOW), which are EBEs with an important lifetime. The total number of UOW resulting from this filtration process was 16, as highlighted in bold font in Table 1.

As suggested by Ould [3], for unseen UOW that may not be derived through heuristics, the author scanned the department names. Moreover, UOW were examined by adding "change to "or "collection of" prior to the UOW, to reveal whether a candidate UOW creates another UOW. Three 'unseen' UOW were identified: blood unit, blood processing, and quality plan.

Deriving UOW from literature is a novel method developed by the authors. First, during the literature review, every term related to blood bank and blood transfusion was examined to see whether it was compatible with the specification of UOW. Subsequently, a list of candidates UOW was developed based on the literature review. Finally, the list was reviewed by both the authors and validated by a domain expert. Consequently, 20 UOW were extracted from the literature.

In comparison to those derived through the Riva method, 19 UOW were identical, in addition to a new UOW called "blood donor recruitment" derived from the literature. Hence, this denotes new information for the candidate process.

**Table 1. Essential business entities and units of work resulted from the questionnaire.** Units of Work (UOW) are highlighted in bold font.

| Accuracy | Data Pool for Research | Irradiation | Infection Screening |
|---|---|---|---|
| **Aphaeresis** | Forward and Reverse blood grouping | Machines Constant Monitoring | Sending Sample to Blood Bank |
| Awareness Lecture | Direct & Indirect Coombs Test | Monitoring for Donation Complication | Security Policies |
| BB&T Protocol | Documentation | O Negative Unit Shortage | Single Donor Platelets Aphaeresis |
| **Biomedical Waste** | Donation Request | Patient Identity Verification | Testing to Other Institution |
| **Blood Component** | Donor Information | Patient Privacy | Threshold to Perform |
| **Blood Component Storage and Distribution** | Drawing Blood | Patient Safety | Timely Connection to Supply Chains |
| **Blood Donation** | Exchange Blood to Sickle Cell Disease | Phenotyping | **Transfusion** |
| **Blood Donation Campaign** | External Audits | Plasma Exchange | Transfusion Time Announced to Technicians |
| Blood Donor Registration | Filtration | Platelet Refractoriness Testing | Transportation |
| Blood Donor Eligibility Check | Following Strict Guidelines | **Post-transfusion** | Validity |
| Blood Donor Outreach | Free Medical Day | **Preparation of Blood Component** | Venesection |
| Blood Double Check | **Funding Facilitation** | Process Mapping | Vital Signs check |
| Blood Group and Cross Match Order | Genotyping | **Quality Control** | Verification |
| Blood Sampling | Hemoglobin Test | Quality of Blood Component | **Blood Shortage Crisis** |
| **Blood Transfusion Request** | Hospital | **Recipient's blood Sample** | Inspection |
| Inspector | Recipient Need | Recipient Information | Cross Matching |
| Client Satisfaction Follow Up | Consent Form | Instructions for Donor | Responsibility |
| Data collection | International Guidelines (AABB, WHO) | Security Policies | International and National Conference |
| **Safety Measure** | | | |

Step 5: Identify Dynamic Relationships Between Units of Work

To design the UOW diagram, the architect ought to determine all the 'generate' relationships among all the UOW that were identified in the previous step [3]. The 'generate' relationship means that if there are two units of work, i.e., one is called A and the other is called B, during the lifetime of UOW A, the UOW B is generated [3]. The 'generate' relationship covers other concepts such as 'activate,' 'require' and "call for" [3]. The UOW diagram is designed where each UOW is represented in a hexagonal shape, and the relationships are indicated by arrows with the titles of the relationships. If an outside relationship is found to generate a UOW, this outside party is represented as a cloud [3].

The 20 UOW extracted using the Riva method and literature review were ordered alphabetically. Each UOW was then selected separately, starting from the first UOW in the list, and its relationships with all the other 19 UOW were examined. The UOW diagram is shown in Fig 2.

Step 6: Transform the Unit of Work Diagram into a First-Cut Process Architecture

To develop the first-cut architecture diagram, three processes are hypothesized for each UOW as follows: the Case Process (CP) occurs during the lifetime of a single case of the UOW [3]. A Case Management Process (CMP) occurs when managing the flow of several cases of the same type of UOW [3]. The Case Strategy Process (CSP) occurs only for UOW that require maintenance or changes [3]. Next, each relationship should be classified into either a 'service' relationship, which means that the generated UOW is generated independently from the generating UOW, or a "task force" relationship, which means the generated UOW is generated dependently on the generating UOW [3]. There are general rules for designing 'service' and "task force" relationships [3].

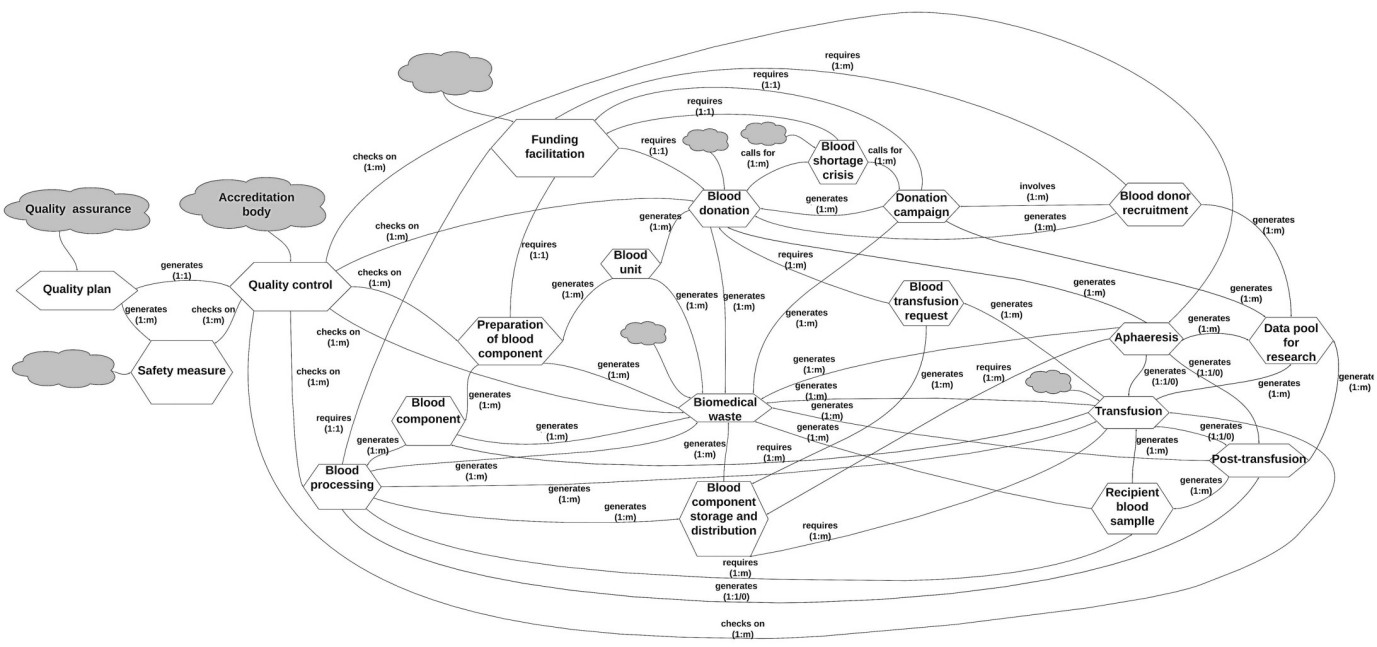

**Fig 2. The Units of Work (UOW) diagram for BB&T organization.**

The general rule for designing the 'task force' relationship states that if UOW (A) generates UOW (B), then (A) CP requests (B) CMP, (B) CMP negotiates (A) CP, (B) CMP starts and monitors, intervenes, or stops (B) CP, (B) CMP negotiates (A) CMP, and (B) CP delivers to (A) CP. The general rule for designing a "service" relationship is similar to the general rule for designing the 'task force' relationship, except that there is no CMP in the service type relationship.

The CPs, CMPs, and CSPs for BB&T organization UOW were determined. According to the domain expert, the UOW that have CSPs are "Funding Facilitation," "Blood Donation Campaign," "Blood Shortage Crisis," "Blood Donation," "Biomedical Waste," "Blood Donor Recruitment," "Data Pool for Research," "Safety Measure," and "Quality Control." Then, all the relationships between the processes of the BB&T organization are classified as "service" relationships or "task force" relationships. The architecture of the first-cut process is shown in Fig 2.

Step 7: Transform the First-Cut Process Architecture into a Second-Cut Process Architecture

In this step, Ould proposed five heuristics to transform the first-cut process architecture into a second-cut process architecture [3]. These heuristics include folding task force CMP into the requesting CP, dealing with 1:1 "generates" relationships, delivery interactions and delivery chains, collections, and empty CMPs [3]. These heuristics were applied to the first-cut BB&T organization diagram to generate the second-cut architecture, as shown in Fig 3, resulting in 45 interrelated processes.

The second-cut process architecture of the BB&T organization is the actual BB&T PA, as shown in Fig 4. This is the first process architecture for the BB&T organization that links all BB&T processes into a coherent blueprint. This can be generalized as a reference process architecture (PA) for all similar BB&T organizations. Moreover, this resultant architecture is a "to-be" model that acts as a BPA blueprint for the business at this stage. Finally, the architecture

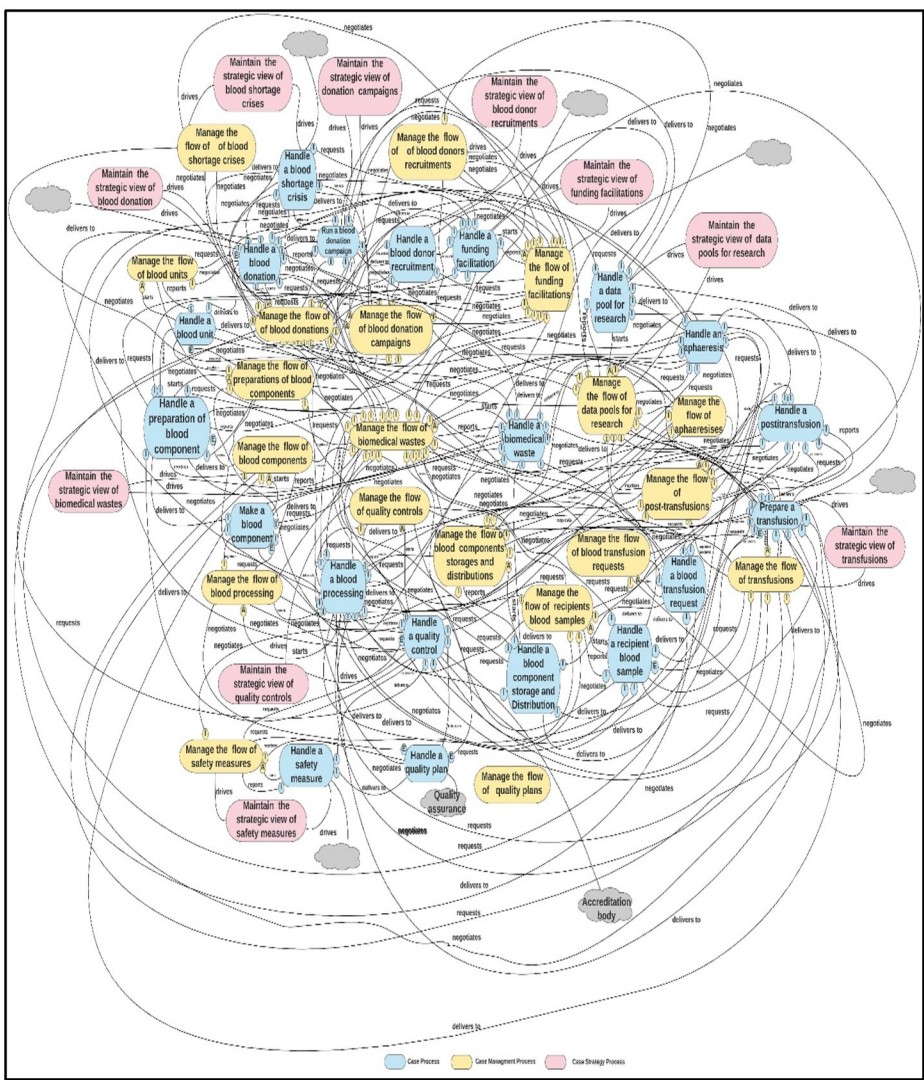

**Fig 3. The first-cut process architecture of the BB&T organization.**

was constructed based on inputs from both the BB&T industry and literature sources to make it as solid, comprehensive, and generic as possible.

Phase 6: Communication

The resulting BB&TPA was communicated to domain experts to evaluate its representativeness. The output solution artifact from these phases, which is the deliverable point of this research, is a highly comprehensive BB&TPA blueprint for virtual organizations.

## Results and discussion

### Evaluating the blood banking and transfusion process architecture

The purpose of this section is to evaluate the resulting BB&TPA by applying it to a real BB&T organization. This was conducted by testing it in the blood bank and transfusion section at KAUH [11, 12]. The processes in the second-cut architecture were compared to real-world

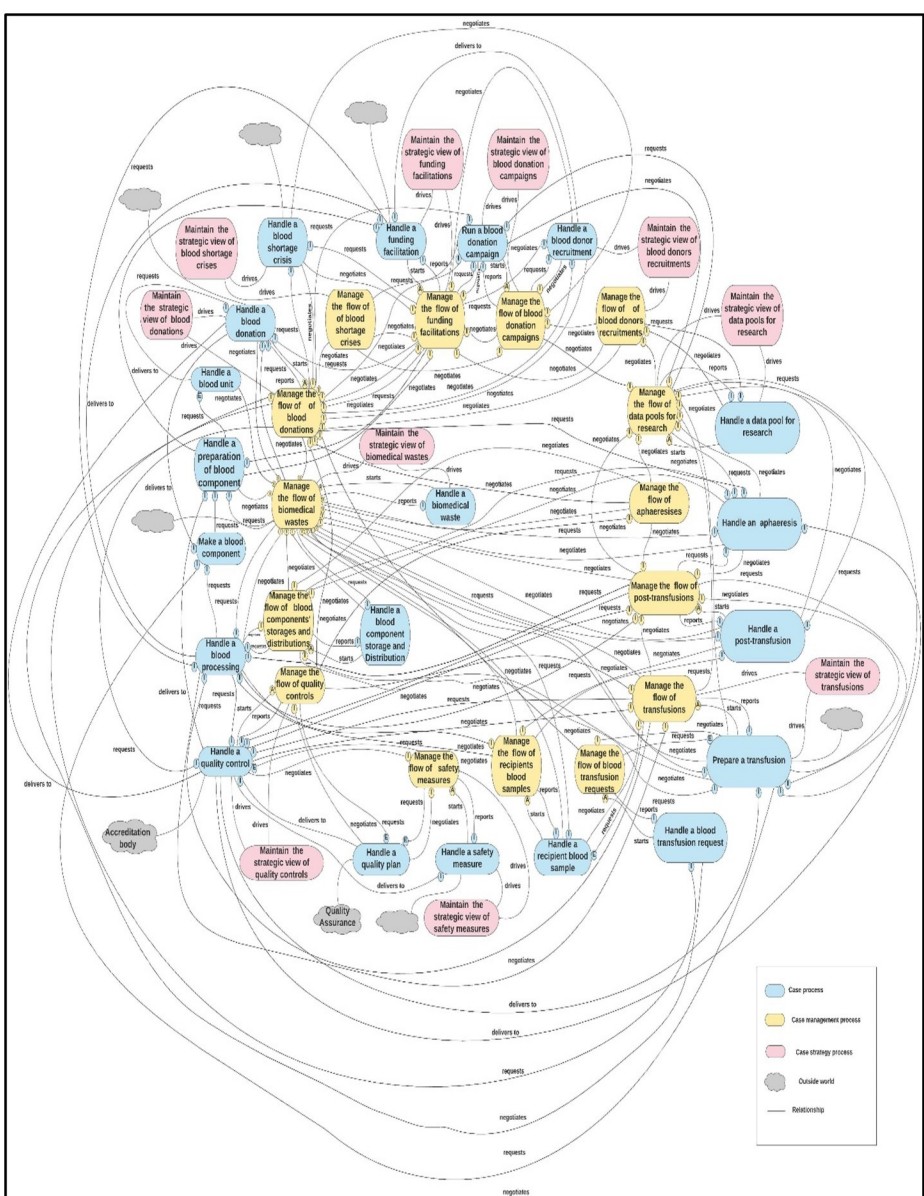

**Fig 4. The second-cut process architecture of the BB&T organization.**

blood banking and transfusion processes. A comparison was performed on several dimensions. The first dimension relates to the process level, verifying whether all processes in the resulting BB&TPA are identified and are running at KAUH. Other comparison dimensions were conducted at the levels of activities, roles, data, Standard Operating Procedures (SOPs), guidelines, and policies for each process.

To manage this evaluation, domain experts divided the second-cut architecture into four function-based units (functional units): "Blood Component," "Blood Donation," "Quality Management," and 'Transfusion' represented as the divisions of the BB&TPA, as shown in Figs 5–8, respectively. Subsequently, a table of five columns demonstrating the evaluation was developed as an evaluation tool. The first column lists the functional unit. The second column

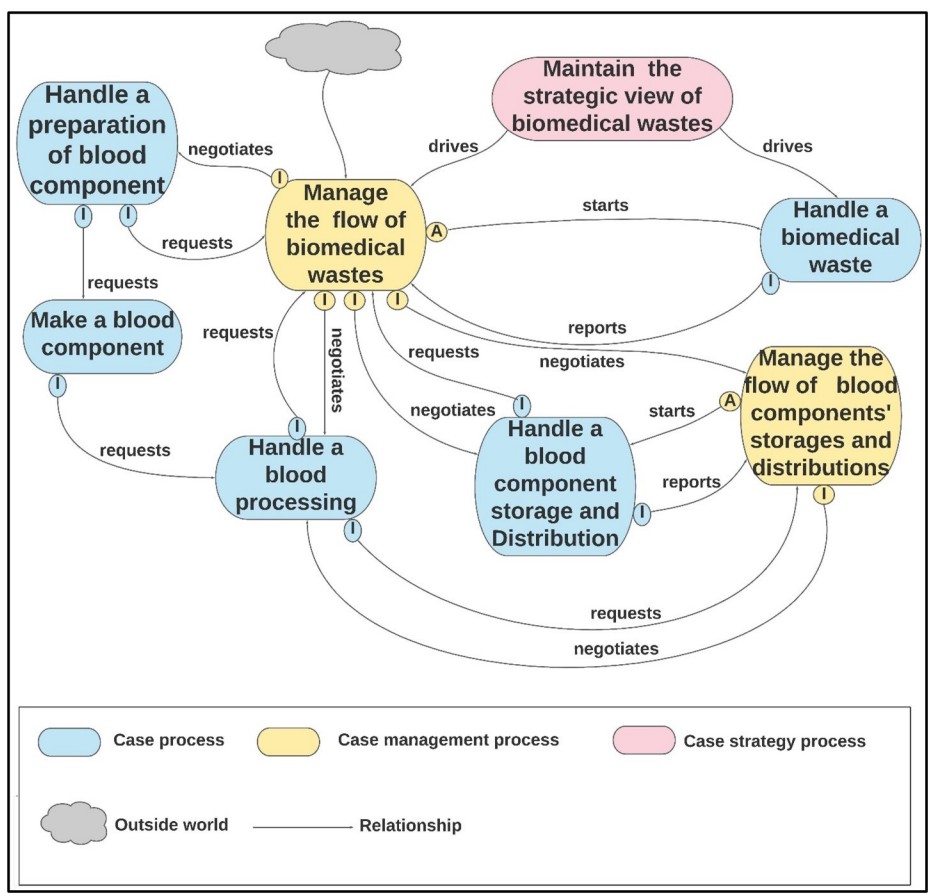

**Fig 5. The second-cut BB &T PA for the 'Blood Component' function unit.**

represents the domain expert agreement regarding the presence of the functional unit at KAUH. The list of all related processes in each unit of BB&T PA is shown in the third column. The fourth column shows whether the processes in BB&T PA currently exist at KAUH. Finally, the fifth column presents the implications of the BB & T PA processes in a real-world example. The last column is divided into sub-columns to address issues related to relationships between processes, roles, data, policies, activities, and SOPs. Table 2 presents the evaluation results.

## Reflections on the blood banking and transfusion process architecture

The work of this paper has made a significant contribution to the BB&T department at KAUH by designing its BPA. The key contributions of the PA diagram are represented in providing a blueprint of the current BB&T processes and their relationships that have already been absent, acting as a common communication ground for stakeholders, and raising awareness among staff about the current running processes and their dependencies, which would facilitate implementing changes starting at a higher level. Most importantly, the development of BPA revealed that 16 BB&T processes were missing at KAUH: three CPs, three CMPs, and all ten CSPs. The term "missing" is defined in two ways: either the process is totally absent in the hospital, and in this case, it is called "absolutely missed process," or the process is formulated generally, not defined specifically, but can be extracted from the hospital or organization's general

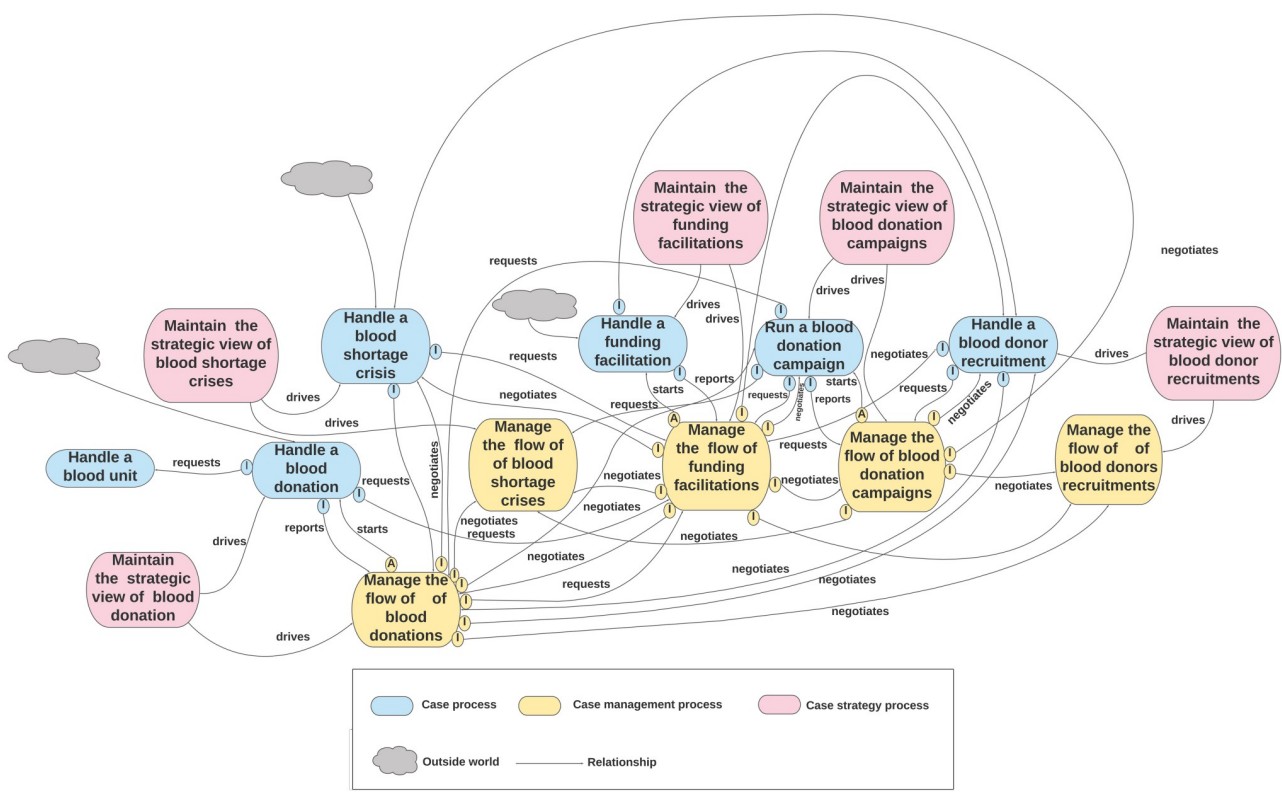

**Fig 6. The second-cut BB&T PA for the 'Blood Donation' function unit.**

processes, and, in this case, it is called "partially missed process." These are mostly related to the strategic view of the blood bank processes; thus, they cannot be defined specifically in the blood bank, and can be extracted from the hospital/organization's strategic view. It is important to highlight these partially missed processes to correlate their dependencies and improve these areas. Table 3 presents the missing BB&T processes at the KAUH.

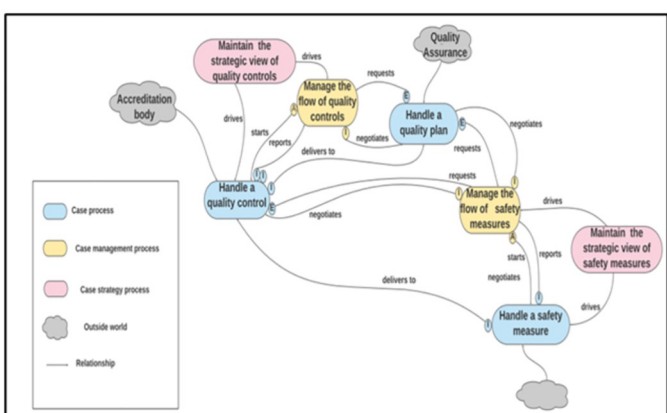

**Fig 7. The second-cut BB&T PA for the 'Quality Management' function unit.**

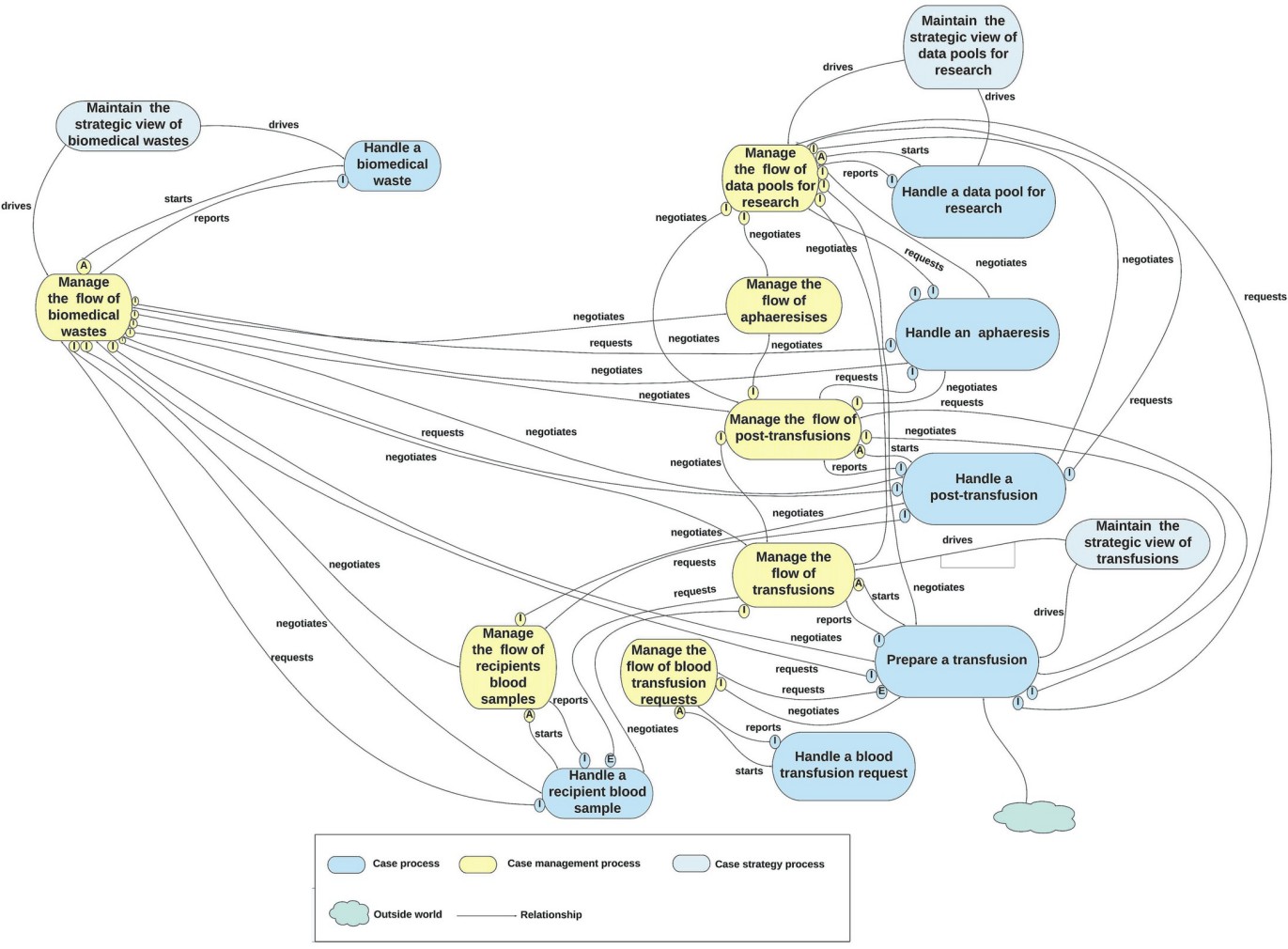

**Fig 8. The second-cut BB&T PA for the 'Transfusion' function unit.**

As shown in Table 3, some processes were absent in all three basic forms (CP, CMP, and CSP), including processes related to "blood donor recruitment," "pool for research," and "funding facilitation." The absence of these processes has implications for other processes. Starting with "blood donor recruitment," the missing CSP ("maintain the strategic view of blood donors' recruitments") leads to a missing written strategic plan for blood donation marketing that aims to raise awareness about the benefits of blood donation through a society that encourages people to donate blood, as well as failing to have a plan to keep blood donors as regular ones. In addition, because of missing the CSP "maintain the strategic view of blood donors' recruitments", both "Manage the flow of blood donors' recruitments" (CMP) and "handle a blood donor recruitment" (CP) will be lost. This means that there is no "handle a blood donor recruitment" (CP) to request "manage the flow of blood donations" (CMP) and "manage the flow of blood donation campaigns" (CMP). Thus, there are no more blood units to replenish the blood stores that usually suffer from shortages. The loss of the negotiation between "manage the flow of blood donors' recruitments" (CMP), which is the process that deals with prioritizing and managing targeted people to be retained as regular blood donors,

**Table 2. Evaluating the BB&TPA in blood bank and transfusion section at KAUH.**

| Function Units in the BB&TPA | Agree on The Presence of Function Units (√/×) | Processes in The BB&T BPA | Processes in The KAUH BB&T business (√/×) | Implications of The Extra Processes in the BB&T BPA on KAUH BB&T Business | | | | |
| --- | --- | --- | --- | --- | --- | --- | --- | --- |
| | | | | Process | Role | Data Produced | Data Consumed | Activity |
| **Blood Donation** | √ | Handle a blood donation | √ | • Outside world<br>• Handle a blood unit<br>• Maintain the strategic view of blood donation<br>• Handle a quality control<br>• Make a blood component<br>• Manage the flow of biomedical waste<br>• Manage the flow of blood donations<br>• Manage the flow of funding facilitations | • Donors<br>• Technician<br>• Donation set (bags, needles, preservatives, and anticoagulation)<br>• Apheresis machine for platelet<br>• Biomedical waste disposal<br>• Blood Transfusion committee | Incident reports<br># of units donated<br># of donors<br>Donor information<br>Microbiology testing (ELISA) statistics | Donation questionnaire (donor data)<br>Donor list | |
| | | Manage the flow of blood donations | √ | • Handle a blood donation<br>• Handle a blood shortage crisis<br>• Run a donation campaign<br>• Manage the flow of blood shortage crises<br>• Manage the flow of funding facilitations<br>• Handle a blood donor recruitment<br>• Manage the flow of blood donors recruitments<br>• Handle an apheresis<br>• Manage the flow of transfusion requests<br>• Manage the flow of quality controls<br>• Manage the flow of biomedical wastes<br>• Handle a quality control<br>• Handle a transfusion request<br>• Maintain the strategic view of blood donations | • Tech. Supervisor<br>• Medical director<br>• Funding rep. | Adequacy of donations | • Inventory status<br>• Crisis information<br>• Funding supplies<br>• Donor demographics | Scheduling donation<br>Prioritizing donor blood groups<br>Negotiating funds Vs needs |
| | | Maintain the strategic view of blood donations | X | • Handle a blood donation<br>• Manage the flow of blood donations | Transfusion committee including clinicians, community rep. BB medical director, national Banks, hospital admin) | Data generated from handling blood | • Hospital strategic view<br>• National blood bank strategic view | • Maintain a safe adequate supply of blood products for patient needs<br>• Insuring donor safety, privacy, and health |

**Table 3. The missing BB&T processes at KAUH.**

| Absolutely Missed BB&T Processes at KAUH | Partially Missed BB&T Processes at KAUH |
| --- | --- |
| **CP** | **CP** |
| Handle a data pool for research | Handle a funding facilitation |
| Handle a blood donor recruitment | |
| **CMP** | **CMP** |
| Manage the flow of data pools for research | Manage the flow of funding facilitations |
| Manage the flow of blood donor recruitments | |
| **CSP** | **CSP** |
| Maintain the strategic view of blood donation campaigns | Maintain the strategic view of blood donations |
| Maintain the strategic view of blood donors recruitments | Maintain the strategic view of funding facilitations |
| Maintain the strategic view of blood shortage crises | Maintain the strategic view of biomedical wastes |
| Maintain the strategic view of safety measures | Maintain the strategic view of quality controls |
| Maintain the strategic view of data pools for research | |
| Maintain the strategic view of transfusions | |

"manage the flow of blood donations" and "manage the flow of blood donation campaigns" at KAUH, is another example of missing relationships between the BB&T processes. The absence of these relationships prevents replenishing the blood bank with the required blood volume.

The "data pool for research" is another significant example of how" basic processes (CP, CMP, and CSP) have been lost at KAUH. The absence of "maintain the strategic view of data pools for research" (CSP), which is responsible for preparing maintenance plans to keep the "manage the flow of data pools for research" (CMP) and "handle a data pool for research" (CP) for ongoing research, will lead to a lack of processes that are driven by it. According to the BB&TPA, "manage the flow of the data pool for research" (CMP) is requested by "prepare a transfusion" (CP), "handle an aphaeresis" (CP), and "handle a post-transfusion" (CP). This is to provide them with the necessary information about the appropriate dose for transfusion and recommendations for the best way to prevent, diagnose, and manage transfusion reactions or complications, such as transfusion-transmitted diseases. Consequently, the absence of "manage the flow of data pool for research" (CMP) means that there are no request and negotiation relationships with "prepare a transfusion" (CP), "handle an aphaeresis" (CP), and "handle a post-transfusion" (CP). This means that there will be no recommendation data to decrease transfusion reactions and other complications; hence, improving "handle a post-transfusion," which is significant to raising the efficiency of "prepare a transfusion" (CP). Furthermore, failing to "handle a data pool for research" (CP), that is, failing to handle one data pool by collecting, analyzing, and representing the data, means there is no "report" relationship to produce a precise report to "manage the flow of the data pool for research" (CMP).

Furthermore, as for the missing basic processes (CSP, CMP, and CP) of "funding facilitation," the absence of "maintain the strategic view of funding facilitations" (CSP) leads to missing both "manage the flow of funding facilitations" (CMP) and "handle a funding facilitation" (CP), because "maintain the strategic view of funding facilitations" (CSP) is responsible for establishing a maintenance plan to ensure the continuous operation of its related CMP and CP. Moreover, the "manage the flow of funding facilitations" (CMP) is required by four CPs: "handle a blood donor recruitment," "handle a blood donation campaign," "handle a blood donation," and "handle a blood shortage crisis," so the absence of "manage the flow of funding facilitations" (CMP) will directly or indirectly reduce the number of "run a donation campaign" (CP), "handle a donation recruitment" (CP) and "handle a blood donation" (CP),

resulting in failure to replenish the volume of donated blood, which will exacerbate the shortage of blood volume, an already existing challenge for blood banks. On the other hand, "manage the flow of funding facilitations" (CMP) is requested by "handle a data pool for research" (CP) which could play a pivotal role in ameliorating the transfusion reaction. Consequently, failure to run the former process may result in failure to achieve the latter. It is highly recommended to look after the basic processes of "funding facilitation" at KAUH for its significant role in supporting several BB&T processes such as "handle a blood donor recruitment" (CP) and "run a blood donation campaign" (CP). This could alleviate the blood volume shortage and transfusion reactions, the main challenges that confront blood banks, transfusion stakeholders, and patients, especially those with cancer.

As mentioned at the beginning of this section, all CSPs appear to be missing at KAUH. Three of these were discussed above, as their related CMPs and CPs were missing. The absence of the other seven CSPs may lead to the absence of several related processes that may affect the sustainability of these critical processes. For example, missing "maintain the strategic view of blood donations" (CSP), "maintain the strategic view of blood donation campaigns" (CSP), and "maintain the strategic view of blood shortage crises" (CSP) may prevent the blood store from further replenishing, which is essential for overcoming the blood shortage. In addition, failing to "maintain the strategic view of safety measures" (CSP), "maintain the strategic view of quality controls (CSP)," "maintain the strategic view of data pools for research" (CSP), and "maintain the strategic view of funding facilitations (CSP)" will have an impact on the quality of the service delivered, such as the incidence and management of transfusion reactions. On the other hand, missing the "maintain the strategic view of biomedical wastes (CSP)" specific to the blood bank would have an impact on environmental safety.

Missing out on CSPs in the process architecture seems to be common, possibly because it is not deemed relevant to the organization or is neglected [21]. Even Martyn Ould, who pioneered the development of the Riva method, did not specify associated activities [22]. Ould stated that CSPs are generally omitted from the process architecture diagram (PAD) unless they are of specific interest [23].

Addressing these missing processes is important for a better understanding of how the BB&T organization performs and runs the business. CSPs, in particular, are crucial for determining future strategies for CPs and CMPs. Furthermore, they are significant in terms of incremental improvements and radical changes. Identifying the missing processes through this research has helped the blood bank at KAUH link the gaps and urged them to rework existing laboratory policies, procedures, and documents at both the departmental and hospital levels, although working on the strategic view of these processes is still in progress [11, 12].

After conducting the research gap analysis, we found that there is no comprehensive blueprint for BB&T processes and their relationships; therefore, one product of this research was developed: the BB&T process architecture comprised 45 processes, as shown in Fig 4. Following the evaluation of BB&TPA by applying it to a real BB&T organization at KAUH, the process architecture was validated to be impactful and was able to inform the correctness and completeness of the blueprint. This blueprint has been used at KAUH to improve BB&T services; one example of the impact of this BB&TPA is identifying "handle a blood donor recruitment" as a missed process, so the blood banking unit is working on adding a policy to facilitate recruitment of donors, especially those who have rare blood types, and also incorporating a mobile phone application to ease the reach to the recurring donors when they are needed based on their location and blood types. It was found that the blueprints of case processes, case management processes, and case strategy processes that have been identified are representative of the domain from a blueprint point-view of processes. The relationships between the processes were found to be correct after they were reviewed. This opens further directions for

research to comprehensively examine the governance of BB&T, and thus, this has also inspired further research that is being developed to do with informing quality governance of BB&T processes through the process architecture. Attempts have been made to build intelligent management systems in BB&T services, but with a fragmented approach, largely focusing on parts of the complete operations, such as the blood donation process [24]. The implementation of BB&T.PArch would serve as a guide for the development of intelligent systems for blood banking and transfusion services in a comprehensive manner.

Finally, it is worth mentioning that the researchers observed shortcomings in the Riva method. Questionnaires were used to answer a set of questions, and the original Riva method does not necessarily suggest using questionnaires. As for strategic modeling, the Riva method also does not have enough guidance; researchers tend to employ the notion of strategic process modeling throughout the transition from the UOW diagram to the first cut and the second cut process architecture. This adds another dimension to the Riva process architecture and its impact on the strategic view of BB&T's virtual organization and opens a new research avenue. Having built the blueprint of processes and their relationships, the next step can be process-modeling of each of those processes.

In terms of study limitations, identifying Essential Business Entities and Units of Work required distinguishing overlapping technical terms, such as "post-transfusion" and "transfusion reaction." Furthermore, despite the comprehensive review of transfusion processes using the Riva method and the literature review search, in which every term related to blood banks and transfusions was examined to see if it was compatible with the UOW specification, additional processes may emerge when the generated BB&TPA is employed in multiple blood centers for some time. The BB&T PArch has been validated in a regional hospital (KAUH) that is accredited by national and international bodies, including JCIA and HCAC (https://www.kauh.edu.jo/Home). Consequently, as this hospital implements and adheres to the generalized services of these accrediting bodies (which adhere to international standards in relation to blood banking and transfusion), the BB&T.PArch is one attempt that portrays a generalized (but not "the" generalized) blueprint of the processes of blood banking and transfusion and their interactions and relationships. However, every implementation of the BB&T PArch is subject to the legal, social, ethical, professional, and technical requirements of the local blood banking and transfusion center, let alone alignment with its healthcare information systems.

## Conclusion

A Riva-based process architecture diagram was successfully developed, showing 45 BB&T interrelated processes that acted as a reference model for a virtual BB&T organization. It is a novel output in the domain of blood banking and transfusion and can also be considered as a reference model to evaluate the existing processes in real-world BB&T units. The DSRM process was adapted for the application of the Riva BPA, which seems to be well-aligned as it led to validating each Riva method step with domain experts. The generated BB&TPA artifact can serve as a knowledge repository for developing a shared understanding of BB&T processes and their linkages. As a result, domain experts can do gap analysis in their BB&T units, plan the development of BB&T management information systems, identify any flaws in their practice, and trace related processes back to the flaw to maximize the impact of improvement, saving time and effort.

Another reflection is the incorporation of these processes in the inspection workflow of accreditation organizations, in which spotting a deficiency in one process can draw attention to processes linked to the deficiency area, working as a map for the inspection team.

The main challenges encountered in this study include the limits of the Riva approach while employing questionnaires and applying strategic modeling. Furthermore, the generated BB&TPA was evaluated in one blood center and additional processes may emerge when it is employed on a wider scale. Further follow- up will be needed.

This work opens further directions for research to comprehensively examine the governance of BB&T, and thus, this has also inspired further research that is being developed to do with informing quality governance of BB&T processes through the process architecture. Finally, as we have developed the blueprint of processes and their relationships, the next stage can be process modeling of each of those processes.

## Supporting information

**S1 Appendix. List of stakeholders' title and number who responded to the research questionnaire.**
(DOCX)

**S2 Appendix. The research questionnaire form.**
(DOCX)

## Author Contributions

**Conceptualization:** Anwar Rjoop, Mohammed Odeh.

**Data curation:** Shaima' Elhaj.

**Formal analysis:** Anwar Rjoop, Shaima' Elhaj, Dina Tbaishat, Yousra Odeh, Mohammed Odeh.

**Investigation:** Anwar Rjoop, Mohammed Odeh.

**Methodology:** Anwar Rjoop, Dina Tbaishat, Yousra Odeh, Mohammed Odeh.

**Project administration:** Anwar Rjoop, Shaima' Elhaj.

**Supervision:** Anwar Rjoop, Dina Tbaishat, Yousra Odeh, Asem Mansour, Mohammed Odeh.

**Validation:** Anwar Rjoop, Shaima' Elhaj.

**Writing – original draft:** Anwar Rjoop, Shaima' Elhaj, Dina Tbaishat, Yousra Odeh, Asem Mansour, Mohammed Odeh.

**Writing – review & editing:** Anwar Rjoop, Shaima' Elhaj, Dina Tbaishat, Yousra Odeh, Asem Mansour, Mohammed Odeh.

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
