## [Decision Letter · Decision Letter 0]

22 Mar 2024

PONE-D-24-00300A Generic Blood Banking and Transfusion Process-Oriented Architecture for Virtual OrganizationsPLOS ONE

Dear Dr. Rjoop,

Thank you for submitting your manuscript to PLOS ONE. After careful consideration, we feel that it has merit but does not fully meet PLOS ONE’s publication criteria as it currently stands. Therefore, we invite you to submit a revised version of the manuscript that addresses the points raised during the review process.

1. Before the resubmission, ensure the manuscript is edited by a first language English speaker and technical words have been thoroughly  edited by your transfusion experts 

2. Ensure the "blood bank" language is technically correct e.g  In line 101, the component is called 'Cryoprecipitate'. Further to this, use blood products not components,   blood products include whole blood

   and components e.g Fresh Frosen Plasma, Cryoprecipitate, platelets etc

3. Line 193: Manage transfusion reaction to post transfusion , 2 different entities , transfusion reactions can be managed both within transfusion and post transfusion

4. Table 1, second row, second column  ( page 6)Direct and indirect blood grouping or  forward and reverse grouping?   Please find out the correct description (refer to point 2 above on technical terms)

5. Lines 322 and 323 :   The absence of these relationships replenishes the blood

bank with the required blood volume.   Please review confirm if the word “ absence” is correct or you meant  “presence”

6. This is to provide them with the necessary information about the appropriate  dose for transfusion and recommendations for the best way to prevent, diagnose, and manage transfusion reactions, such as platelet refractoriness and transfusion-transmitted diseases. Consequently, the absence of “manage the flow of data pool for research” (CMP) means that there are no request and negotiation relationships with “prepare a transfusion” (CP), “handle an aphaeresis” (CP)

336 and “handle a post-transfusion” (CP).   Please review “ platelet refractoriness”, this has nothing to do with transfusion reaction, its under blood donation. May be this indicate the limitation challenges of the process.

7. Line  367, replace “bereave” with a better common word

8. Rephrase lines 372 to 375 ,

9. 406 to 414, also include limitations of this study  as per comment on line 336

10. Line 415: Any limitations of the study ? again in reference to comment on line 336 and other limitations of RIVA method in general

11. Conclusion should highlight challenges/limitations of  the study, and also avoid generalization ( i.e this study was done only in one blood bank. We do not know know if the results will e the same in other blood banks.   

We look forward to receiving your revised manuscript.

Kind regards,

Stephen Emilio Njolomole, MB,BS ,MPH

Guest Editor

PLOS ONE

4. We notice that your supplementary figures are uploaded with the file type 'Figure'. Please amend the file type to 'Supporting Information'. Please ensure that each Supporting Information file has a legend listed in the manuscript after the references list.

5. We notice that your supplementary tables are included in the manuscript file. Please remove them and upload them with the file type 'Supporting Information'. Please ensure that each Supporting Information file has a legend listed in the manuscript after the references list.

Reviewers' comments:

Reviewer's Responses to Questions

**Comments to the Author**

1. Is the manuscript technically sound, and do the data support the conclusions?

Reviewer #1: Partly

Reviewer #2: Yes

Reviewer #3: Partly

2. Has the statistical analysis been performed appropriately and rigorously? 

Reviewer #1: I Don't Know

Reviewer #2: N/A

Reviewer #3: I Don't Know

3. Have the authors made all data underlying the findings in their manuscript fully available?

Reviewer #1: Yes

Reviewer #2: Yes

Reviewer #3: No

4. Is the manuscript presented in an intelligible fashion and written in standard English?

Reviewer #1: No

Reviewer #2: Yes

Reviewer #3: No

5. Review Comments to the Author

Reviewer #1: At first, I must appreciate the effort and newness you put into this manuscript. But the problem is that it is not written in a simple language that is easy to understand. The abstract should be more precise. In line 101, the component is called 'Cryoprecipitate'.

Reviewer #2: Manuscript is presented in an intelligible fashion and written in standard English. Figures depicted by authors, contain large pool of data , make the figures complicated .It require better representation.

Reviewer #3: The article titled " A Generic Blood Banking and Transfusion Process-Oriented Architecture for Virtual Organizations " has been reviewed in detail by me. My comments are below;

1. As far as it is understood in the research, a management system has been designed on a hospital basis through a control program that controls the entire process of blood banking from the recruitment of volunteer donors to the follow-up of reactions arising from the transfusion of the blood product reaching the patient and to alert managers by identifying missing or inadequate steps in the process. However, how the system works and how it will contribute to the practice is not explained in a language that is understandable enough for blood center managers and employees.

2. The article proposal creates the impression of a report with a lot of technical details.

3. It would be much better if this management system is actively used in one or a few blood centers for a long period of time and the positive contributions that emerge after the follow-up of its reflection on practice are explained in an understandable way. As it is, the method and results of the study were not found to be sufficiently comprehensible by the audience.

6. PLOS authors have the option to publish the peer review history of their article (what does this mean?). If published, this will include your full peer review and any attached files.

Reviewer #1: **Yes: **Sushanta Kumar Basak

Reviewer #2: No

Reviewer #3: No

---

## [Author Response · Author response to Decision Letter 0]

27 Apr 2024

Dear Editor/reviewers, 

Thank you for these comments designed to improve our paper, “A Generic Blood Banking and Transfusion Process-Oriented Architecture for Virtual Organizations,” which we have addressed below. We greatly appreciate the time and effort put forth by reviewers and editors to improve our paper. If any response is unclear or you wish for additional changes, please let us know. Please note that we submitted a file with tracked changes (review reports) and a clean copy of the revised manuscript. The line and page numbers in the responses match the file's tracked modifications.

Sincerely,

Anwar Rjoop

ensure the manuscript is edited by a first language English speaker and technical words have been thoroughly edited by your transfusion experts The manuscript is edited for grammatical and typographical errors and a transfusion expert reviewed it. However, we will gladly review any unseen faults.

Ensure the "blood bank" language is technically correct e.g In line 101, the component is called 'Cryoprecipitate' Thank you for your comment. The term has been corrected to cryoprecipitate.

Line 103, page 5.

3. Line 193: Manage transfusion reaction to post transfusion, 2 different entities, transfusion reactions can be managed both within transfusion and post transfusion Thank you for highlighting this issue. As we described the method of extracting the processes using the RIVA object-based process architectural modeling method supported by systematic mapping of the literature. Some of the Riva EBEs (Essential Business Entities) and in particular the ones from managing transfusion to its post transfusions were renamed to encompass the relevant as many processes. We opted to use “post transfusion” for all processes encountered after starting the transfusion, even those occurring within the transfusion. This is clarified in line 207, page 10.

4. Table 1, second row, second column (page 6)Direct and indirect blood grouping or forward and reverse grouping? Please find out the correct description (refer to point 2 above on technical terms) Thank you for your comment. There are two distinct parts to ABO grouping. The Direct or Forward grouping requires known anti-A and anti-B typing antiserums for testing unknown cells. The Indirect, Reverse or Back grouping requires a pool of known group A and B cells. As these terms may be used interchangeably, the terms Forward and Reverse might be more common. Hence, we changed the term in table 1 to Forward and Reverse blood grouping.

5. Lines 322 and 323: The absence of these relationships replenishes the blood

bank with the required blood volume. Please review confirm if the word “ absence” is correct or you meant “presence” Thank you for your comment aiming to improve the article. The sentence is reformatted to (The absence of these relationships prevents replenishing the blood bank with the required blood volume).

Line 351, page 19 

6. This is to provide them with the necessary information about the appropriate dose for transfusion and recommendations for the best way to prevent, diagnose, and manage transfusion reactions, such as platelet refractoriness and transfusion-transmitted diseases. Consequently, the absence of “manage the flow of data pool for research” (CMP) means that there are no request and negotiation relationships with “prepare a transfusion” (CP), “handle an aphaeresis” (CP)

336 and “handle a post-transfusion” (CP). Please review “platelet refractoriness”, this has nothing to do with transfusion reaction, it’s under blood donation. May be this indicate the limitation challenges of the process. Thank you for your comment. We agree that platelet refractoriness is not under transfusion reaction. The examples were meant to refer to the processes in the lines above. However, to better formulate the sentence, platelet refractoriness is removed.

Line 363, page 20.

7. Line 367, replace “bereave” with a better common word Thank you for your comment. The word bereave is replaced with “prevent”. 

Line 397, page 21 and 22

8. Rephrase lines 372 to 375, Thank you for your comment. The lines are rephrased.

Lines 401-405, page 17

9. 406 to 414, also include limitations of this study as per comment on line 336 Thank you for this valuable feedback. These limitations are discussed. 

Lines 455–468, page 24

10. Line 415: Any limitations of the study? again in reference to comment on line 336 and other limitations of RIVA method in general Thank you for this valuable feedback. These limitations are also discussed in reference to the previous comment (9).

Lines 455–468, page 24

11. Conclusion should highlight challenges/limitations of the study, and also avoid generalization ( i.e. this study was done only in one blood bank. We do not know if the results will be the same in other blood banks. Thank you for your comment. The challenges and limitations of the study are added to the conclusion.

Lines 494-497, page 25

Please ensure that your manuscript meets PLOS ONE's style requirements The manuscript is edited to meet PLOS ONE’s style.

We note that your Data Availability Statement is currently as follows: [All relevant data are within the manuscript and its Supporting Information files.]

Please confirm at this time whether or not your submission contains all raw data required to replicate the results of your study

 Yes, the submission and supplementary material contain all the data required to replicate the results of the study

3. Your ethics statement should only appear in the Methods section of your manuscript The ethic statement is moved to Methods section

4. We notice that your supplementary figures are uploaded with the file type 'Figure'. Please amend the file type to 'Supporting Information'. Please ensure that each Supporting Information file has a legend listed in the manuscript after the references list. Figures and tables are submitted to file-type 

‘Supplementary information’.

Figures 1-8 and Tables 1-3 are part of the manuscript. However, File 1 and File 2 are supplementary and are added after reviewer comments.

Please let me know if further modifications are needed.

5. We notice that your supplementary tables are included in the manuscript file. Please remove them and upload them with the file type 'Supporting Information'. Please ensure that each Supporting Information file has a legend listed in the manuscript after the references list. Tables are removed from the manuscript and uploaded with the file type ‘Supplementary information’.

Legends are listed in the manuscript as requested. 

6. Please review your reference list to ensure that it is complete and correct. If you have cited papers that have been retracted, please include the rationale for doing so in the manuscript text, or remove these references and replace them with relevant current references. The reference list has been reviewed as requested. No retracted articles were found.

An additional reference [24] is added in response to reviewer#3 comment

Attached file comments:

1. In methods and design, authors stated the involvement of 20 key stakeholders. Who were the stakeholders and what was the criteria for selection of them?

 Thank you for the comment. A stakeholder definition is added (line183-184, page 9). A stakeholder list including the number and description of each person who responded to the questionnaire, is included in the supplementary material along with the questionnaire form.

2. No clarity of response is earmarked in column four of table 2 about availability of process at KAUH Thank you for highlighting this note. The response is added to table 2, column 4. Extra columns are removed from the table (the last 3 columns on the right) since they have no additional data to present.

5. Review Comments to the Author

Reviewer #1: At first, I must appreciate the effort and newness you put into this manuscript. But the problem is that it is not written in a simple language that is easy to understand. The abstract should be more precise. In line 101, the component is called 'Cryoprecipitate'.

Response: 

Thank you for your comment aiming to improve the article. The manuscript is indeed technical because of the nature of the subject, which requires technical terms from a blood bank and informatics point of view. We have now further tried in this revision to simplify the language as much as possible without compromising the health informatics scientific content of the article. For example, see lines 476-480 in page 25. 

The abstract is rephrased to be more concise.

Cryoprecipitate: Thank you for your comment. The term has been corrected.

Line 103, page 5.

Reviewer #2: Manuscript is presented in an intelligible fashion and written in standard English. Figures depicted by authors, contain large pool of data, make the figures complicated. It requires better representation.

Response: 

Thank you for your comment aiming to improve the article. We agree that the figures contain a large pool of data, which is inherent to the subject. We tried to explain the process of generating these figures in the text. Additionally, the second-cut architecture, which is the blueprint of the process architecture, is divided into four functional units that are presented in separate figures for this purpose. 

Reviewer #3: The article titled " A Generic Blood Banking and Transfusion Process-Oriented Architecture for Virtual Organizations " has been reviewed in detail by me. My comments are below;

1. As far as it is understood in the research, a management system has been designed on a hospital basis through a control program that controls the entire process of blood banking from the recruitment of volunteer donors to the follow-up of reactions arising from the transfusion of the blood product reaching the patient and to alert managers by identifying missing or inadequate steps in the process. However, how the system works and how it will contribute to the practice is not explained in a language that is understandable enough for blood center managers and employees.

Response: 

Thank you for your comment aiming to improve this article. The manuscript indeed describes the development of a generic blueprint of a business process architecture unique to blood bank and transfusion (BB&T PArch) services utilizing the state-of-the-art Riva Process Architecture modeling methodology. This has led to the identification of a highly comprehensive set of processes that are anticipated to exist in any blood banking and transfusion center or service. However, the automation of the BB&T PArch is so subjective for its alignment with the blood banking and transfusion center information systems that they themselves may be siloed or islands of information not talking to each other. This latter explanation is the subject of further research on bridging the digital divide in this context. However, we have demonstrated the effectiveness of the BB&T PArch via its validation at a regional hospital, namely, KAUH, which impacted the service positively. One example of the impact of this BB&T PA is identifying “handle a blood donor recruitment” as a missed process, so the blood banking unit is working on adding a policy to facilitate the recruitment of donors, especially those who have rare blood types, and also incorporating a mobile phone application to ease the reach of recurring donors when they are needed based on their location and blood types. We have now further tried in this revision to simplify the language as much as possible without compromising the health informatics scientific content of the article. For example, see lines 476-480 in page 25. 

2. The article proposal creates the impression of a report with a lot of technical details.

Response: 

Thank you for your comment. Every effort has been attempted to revise and simplify the technical language in the article; however, the nature of the subject process architecture modeling dictates some level of newly introduced technical terms that every attempt has been taken to harness unfamiliarity and come close to the language of a process-oriented view of undertaking the interaction of blood banking and transfusion services contextually within the related services provisioning.

3. It would be much better if this management system is actively used in one or a few blood centers for a long period of time and the positive contributions that emerge after the follow-up of its reflection on practice are explained in an understandable way. As it is, the method and results of the study were not found to be sufficiently comprehensible by the audience.

Response: 

Thank you for the valued comment. This could certainly be the case had the aim been to implement this as a management information system for automating blood banking and transfusion services. However, the BB&T PArch is geared towards the development of a generalized process architecture for the domain of blood banking and transfusion. BB&T PArch has been developed using the Riva process architecture modeling method, where domain specific architectures developed using this method are conjectured to be reproducible in organizations in the same business, for example, in our research case, the domain of blood banking and transfusion. In literature, attempts have been made to build intelligent management systems in BB&T services, but with a fragmented approach, largely focusing on parts of the complete operations, such as the blood donation process [24]. The implementation of BB&T PArch would serve as a guide for the development of intelligent systems for blood banking and transfusion services in a comprehensive manner.

Furthermore, the BB&T PArch has been validated in a regional hospital (KAUH) that is accredited by national and international bodies, including JCIA and HCAC (https://www.kauh.edu.jo/Home). Consequently, as this hospital implements and adheres to the generalized services of JCIA (which adhere to international standards in relation to blood banking and transfusion), the BB&T PArch is one attempt that portrays a generalized (but not “the” generalized) blueprint of the processes of blood banking and transfusion and their interactions and relationships. However, every implementation of the BB&T PArch is subject to the legal, social, ethical, professional, and technical requirements of the local blood banking and transfusion center, let alone alignment with its healthcare information systems. 

In conclusion, this research paves the way for further advanced attempts to bridge the divide between developing a highly representative blueprint of blood banking and transfusion processes and their enactment in the respective information systems of the designated healthcare entities for automating the data capture and intelligent processing of data consumed and produced by the concerned blood banking and transfusion services. Thus, this paves the grounds for future research into descriptive and predictive data analytics for blood banking and transfusion services. Lines 461-468 page 24.

---

## [Editor Report · Decision Letter 1]

6 May 2024

A Generic Blood Banking and Transfusion Process-Oriented Architecture for Virtual Organizations

PONE-D-24-00300R1

Dear Dr. Rjoop,

We’re pleased to inform you that your manuscript has been judged scientifically suitable for publication and will be formally accepted for publication once it meets all outstanding technical requirements.

Kind regards,

Stephen Emilio Njolomole, MB,BS ,MPH

Guest Editor

PLOS ONE
---

## [Editor Report · Acceptance letter]

10 May 2024

PONE-D-24-00300R1 

PLOS ONE

Dear Dr. Rjoop, 

I'm pleased to inform you that your manuscript has been deemed suitable for publication in PLOS ONE. Congratulations! Your manuscript is now being handed over to our production team.

Kind regards, 

on behalf of

Dr. Stephen Emilio Njolomole 

Guest Editor

PLOS ONE